Original research

# Factors associated with positive user experience with primary healthcare providers in Mexico: a multilevel modelling approach using national cross-sectional data

Kelsey Holt [iD] ,[1] Svetlana V Doubova [iD] ,[2] Dennis Lee,[3] Ricardo Perez-Cuevas,[4] Hannah H Leslie [iD] [5]

For numbered affiliations see end of article.

**Correspondence to**
Dr Kelsey Holt;
kelsey.holt@ucsf.edu

## ABSTRACT

**Objective** This study aimed to investigate factors associated with patient experience with primary care in a large public health system in Mexico and determine the amount of variability in experience attributable to facility-level and state-level factors.

**Methods** We analysed cross-sectional 2016 national satisfaction survey data from the Mexican Social Security Institute (IMSS). Patient-level data were merged with facility-level data and information on poverty by state. We assessed general contextual effects and examined the relationship of patient, facility and state factors with four patient experience measures using random effects logistic regression.

**Results** 25 745 patients' responses from 319 facilities were analysed. The majority experienced good communication (78%), the opportunity to share health concerns (91%) and resolution of doubts (85%). 29% of visits were rated as excellent. Differences between facilities and states accounted for up to 12% and 6% of the variation in patient experience, respectively. Inclusion of facility-level contextual effects improved model predictions by 8%–12%; models with facility random effects and individual covariates correctly predicted 64%–71% of individual outcomes. In adjusted models, larger patient population was correlated with worse reported communication, less opportunity to share concerns and less resolution of doubts. Men reported more positive communication; older individuals reported more positive communication and experiences overall, but less opportunity to share concerns; and more educated individuals were less likely to report positive communication but more likely to report resolution of doubts and overall positive experiences. Preventive care visits were rated higher than curative visits for resolution of doubts, but lower for opportunity to share concerns, and specific conditions were associated with better or worse reported experiences in some cases.

**Conclusion** Quality improvement efforts at IMSS facilities might bolster individual experiences with primary care, given that up to 12% of the variation in experience was attributable to facility-level differences. The relationship between individual characteristics and experience ratings

## Strengths and limitations of this study

- ► This is one of very few studies to examine factors associated with positive user experience with primary care in a representative sample from a middle-income country.
- ► This study uses multilevel modelling techniques to examine the variance in user experience attributable to factors at the individual versus facility or state levels.
- ► Data are cross-sectional and thus we are unable to draw causal inferences from the observed associations.
- ► Findings are not generalisable to other sectors of the Mexican healthcare system, outside of the Mexican Institute of Social Security.

reinforces the importance of patients' expectations of care and the potential for differential treatment by providers to impact experience.

## INTRODUCTION

High-quality primary care is essential to countries' ability to achieve Universal Health Coverage by 2030, one of the United Nations Sustainable Development Goals (SDGs) established in 2015.[1 2] The core values of primary care, including person-centredness and health equity, are in line with the SDGs' orientation towards reducing inequities and achieving good health and well-being for all[1 3]: primary care provides a platform for delivering care that meets the entire populations' needs.[2]

In order to ensure that primary care is high quality and able to achieve the promise of promoting health for all, monitoring patient experiences is critical. Positive user experience is a core component of high-quality healthcare and contributes to the population's

confidence in the health system and achieving better health outcomes.[4] For instance, positive user experiences with primary care have been found to be associated with public perception of high healthcare quality,[5] self-rated health[6] and high patient satisfaction.[7] Reviews of patient experiences in both primary and secondary care have found associations with clinical effectiveness and patient safety[8] and patient satisfaction.[9]

User experiences are the products of different patient and health services-related factors. Research examining the impact of health services-related factors on patient experience of care must account for the hierarchical nature of data: patients nested within providers, nested within institutions, nested within geographies.[10] Some studies, primarily from high-income countries, have examined the association between factors at different levels and patient experiences. At the individual level, age, education and health status are the most common characteristics associated with patient experiences, with older and healthier individuals, and those with less education, tending to report more positive experiences.[11] Further, expectations for care play a clear role in overall satisfaction with care such that low expectations are associated with higher satisfaction.[9] Patient experience measures are increasingly recognised as valuable for quality improvement purposes as they are known to discriminate more effectively between practices than patient satisfaction measures.[4 12] However, low expectations likely still influence to some degree the ways in which individuals report experiencing care.

At the health services level, variables such as presence of specific types of providers (eg, nurses, in addition to physicians), health facility structure (eg, fewer number of patients per provider, fewer numbers of physicians per practice, comprehensiveness of services provided by physicians) and functioning (eg, long weekly working hours, non-training vs training facilities, payment structure and operational agreements with other health systems) have been found to be associated with positive user experience with primary care.[13–16]

Much of this research is based in high-income settings. Similar research in lower-resource settings would provide crucial information to guide decision-makers in how to allocate resources to address deficiencies in system responsiveness to patients, and to the public to empower them to demand more patient-centred care. Further, in order to understand the degree to which experience measures are reliable indicators of the performance of healthcare systems (rather than a more limited indicator of variability in provider–patient relationships by individual-level sociodemographic factors), it is critical to conduct analyses on the variability of patient experience attributable to health services-related factors.

Consistent with global initiatives, Mexico has a strong national commitment to both primary care and quality improvement to ensure the promises of universal access are achieved.[17 18] Since 2009, user satisfaction and experience surveys are regular practice for public health

providers in Mexico. The Mexican Institute of Social Security (IMSS) is the largest provider, covering more than 63 million formal labour market employees and their families.[19] IMSS conducts annual National Satisfaction Surveys; yet, their results have not been analysed comprehensively.

In this paper, we take advantage of this robust data collection effort by IMSS to investigate the degree to which variability in patient experience is due to clustering at facility and state levels (general contextual effects) and the patient, facility and geographic factors influencing experience with healthcare in the IMSS primary care system (ie, individual level and specific contextual effects).

## METHODS
### Data sources
Our primary dataset comes from the November 2016 IMSS National Satisfaction Survey (ENSAT). IMSS regularly conducts these cross-sectional surveys with its beneficiaries over 18 years old at primary, secondary and tertiary care services. This analysis focuses on patients at the primary care level. The survey's sampling design consists of a two-stage stratified probabilistic sampling where the probability of selecting a health facility is defined by region and facility size. The number of patients to be interviewed in each facility is estimated according to the average number of daily consultations with physicians and the proportion of patients satisfied in the previous survey. Patient recruitment is done through systematic sampling, with a random starting point and a fixed interval based on the estimated sample size and the average number of daily consultations in the clinic. A private firm (Berumen and Associates) conducted the survey. Trained interviewers carried out direct, structured interviews with IMSS health services users. The interviewers used a satisfaction questionnaire that the Centre for Evaluation Research and Surveys of the National Institute of Public Health had previously validated. The response rate in the survey was 87%.

To perform the analysis, we merged the 2016 ENSAT database with a facility-level database of a 2016 physical inventory of healthcare facilities and a 2016 record of patients affiliated with IMSS in each facility. Finally, a 2014 database of the National Council for the Evaluation of Social Development Policy (Spanish acronym CONEVAL) on the percentage of the population in each state living in poverty was merged into patient-level data.[20] Poverty is calculated by CONEVAL as a multidimensional measure based on criteria related to housing, household services, education, healthcare, social security, food and per capita income.

### Patient experience outcomes
We examined four binary dependent variables. The first is a composite measure of verbal and non-verbal communication, designated as positive if the patient answered favourably to three binary questions (translated from Spanish): "Did the staff who attended to you make eye

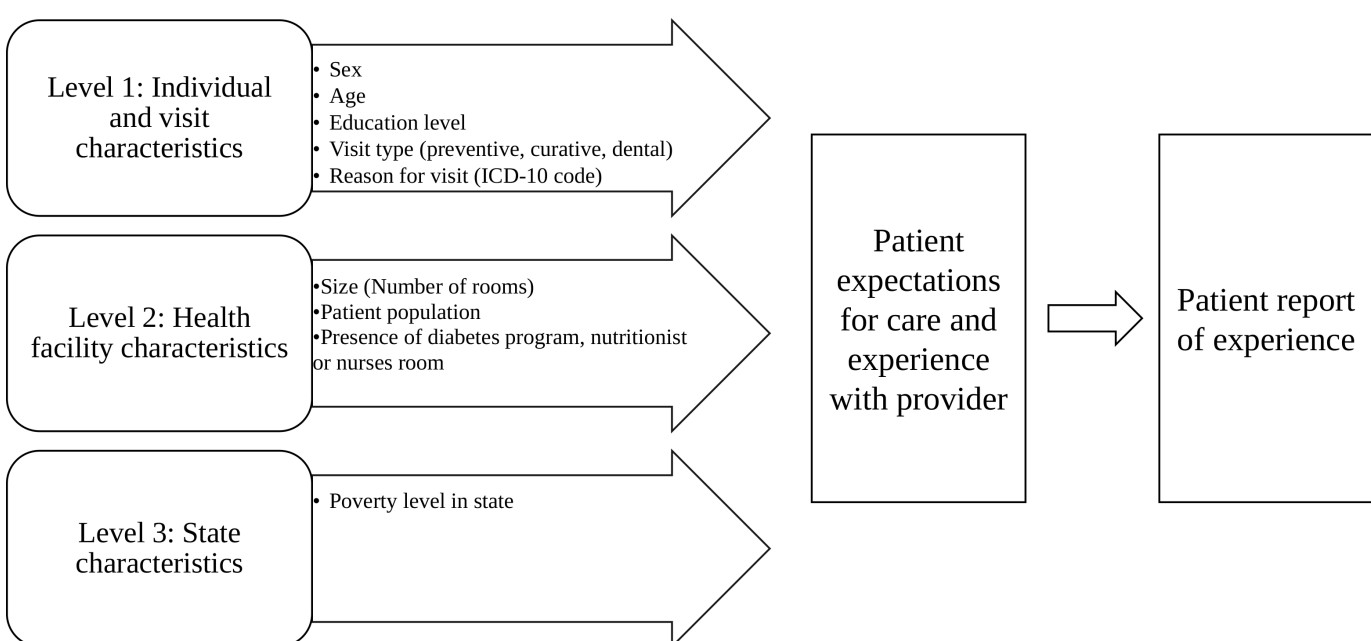

**Figure 1** Conceptual model of relationship between model variables and reported experience with provider (N=25 745 patients and 319 facilities).

contact while greeting you?"; "Did the staff who attended to you listen to you with attention and without interruptions?"; and "Did the staff who attended to you respond clearly to your questions?" We also examined the questions: "Did you have the opportunity to tell the doctor or nurse what was bothering you about your health?" (referred to hereafter as 'opportunity to share health concerns'); "Did you end up with doubts about the treatments necessary for your condition?" (reverse coded and referred to hereafter as 'resolution of doubts'); and "How was the treatment you received in this facility during today's visit?", scored on a scale of 1–5, with 1 being 'excellent,' and 5 being 'terrible,' collapsed to excellent versus all other responses and referred to hereafter as the global measure.

### Explanatory variables

In identifying independent variables to include in our models, we conceptualised patient experiences with IMSS healthcare delivery as being potentially influenced by factors at multiple levels (figure 1). Available individual-level variables in the IMSS dataset included age (categorised into quartiles); sex (binary); and education level (high school or higher, less than high school, and less than primary school). We hypothesised that patients' age, sex and educational status will explain some variance in reported experience due to differential treatment by providers and different expectations for care based on these factors. We also hypothesised experiences would vary by the primary reason for the visit, independent of patient characteristics, given that the nature of what is covered in a visit likely influences how providers approach patients (eg, primary care doctors are less comfortable discussing stigmatised issues such as sexual health and therefore exhibit poorer communication

skills) and how well the visit can meet the patient's needs (eg, if their issue has numerous treatment options, they may have a better experience than for those issues where treatments are limited). We therefore included the variables of self-reported reason for visit, collapsed from 149 categories into 10 categories based on International Classification of Diseases, Tenth Revision (ICD-10) codes, and type of consultation (curative visit, preventive care visit at the PREVENIMSS module at each clinic,[21] or dental visit). PREVENIMSS is a Spanish acronym for 'Integrated Preventive Care Programme,' an initiative that provides targeted preventive care services to individuals based on their age and gender.

At the facility level, we hypothesised that size of facility could impact processes of care and included two variables that served as proxies for size: total number of rooms for all types of services and the number of people affiliated with a primary care clinic. We also included variables related to the presence or absence of programmes that are designed to improve quality of care: presence of a special diabetes programme comprising multidisciplinary teams of health-professionals (DIABETIMSS); presence of nurses' rooms attached to family doctors' consultation rooms; and availability of a nutritionist. Finally, we hypothesised that experience would vary by the percentage of population in a state that is living in poverty, given that this is likely reflective of the infrastructure in the state—including healthcare infrastructure. We thus included poverty rate in the facility's state as an independent variable. Continuous variables (consultation rooms, number of affiliates, poverty rate) were rescaled to fall on a scale of minimum 0 to maximum 1 to facilitate comparison across explanatory variables.

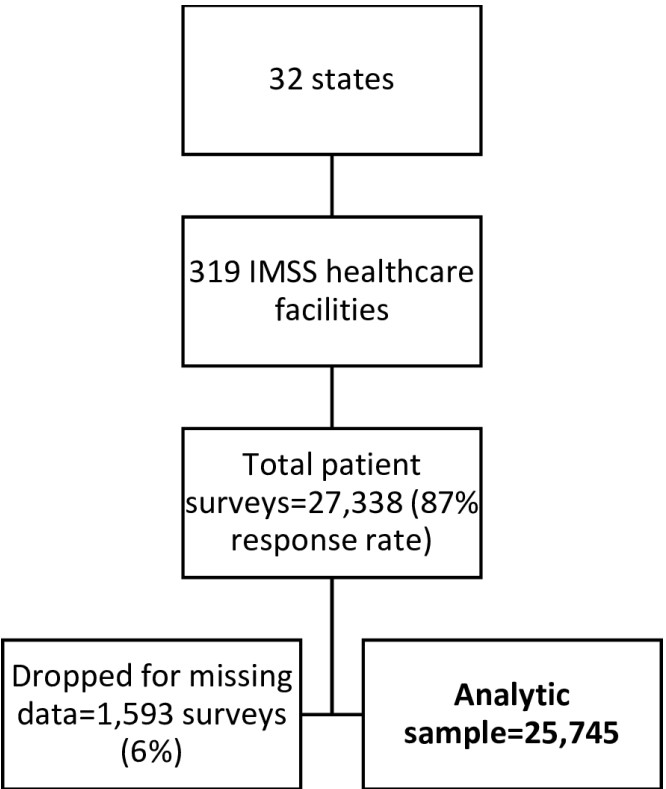

32 states

319 IMSS healthcare facilities

Total patient surveys=27,338 (87% response rate)

Dropped for missing data=1,593 surveys (6%)

Analytic sample=25,745

**Figure 2** Sample flow chart. *Average patients per facility=81 (min=28; max=139); average facilities per state=10 (min 5, max 23). IMSS, Mexican Institute of Social Security.

## Analysis

We performed a series of analyses to address questions about the extent of clustering in each outcome (general contextual effects) and the associations of state and facility factors with each outcome (specific contextual effects).[22] We first constructed a conventional logistic regression model for each dependent variable, including all patient-level explanatory variables of interest (Model 1). We next added a random intercept for health facility (Model 2) and subsequently the facility-level factors that could explain the measured outcomes (Model 3). In the last set of models, we added a state-level random effect (Model 4) and then state-level poverty (Model 5).

For each model, we calculated the area under the receiver operating characteristic curve (AUC). The AUC is a measure of discriminatory accuracy, that is, how well a given model predicts individual outcomes. Comparing the AUC across models as contextual effects are added supports assessment of how the addition of each random effect improves the model's predictive capability. For random intercept models, we calculated the intraclass correlation (ICC) to determine the amount of the total variance in the outcome due to differences at the facility level (Model 2) and the facility and state levels (Model 4).

For specific contextual effects, we calculated the proportional change in variance (PCV) values between steps to allow for examination of the proportion of the contextual variance that is explained by the facility-level and then state-level covariates included in the model. To

interpret the association of facility-level and state-level factors with the outcomes, we calculated population average ORs. Random intercept models allow for inference on the within-cluster difference associated with each covariate. Population average ORs allow for comparing subjects from different clusters (ie, states) that have identical observed values for other covariates. We also examined interval ORs (IORs) at facility and state levels to compare the magnitude of each higher-level covariate to the random effect. For instance, a facility IOR provides the range of ORs comparing individuals with a one unit difference on the facility or state covariate from the 10th percentile to the 90th percentile of the random effect.[23 24]

The IMSS dataset included weights to account for non-response. To account for missing data, we weighted complete cases based on inverse probability weights.[25] Weights were calculated as the inverse of the probability of being a complete case predicted from regressing a binary indicator of completeness on patient-level covariates with no missing values (all except reason for visit and education level). The characteristics of the incomplete cases and the distribution of the inverse probability weights are shown in online supplementary table 1 and online supplementary figure 1; all measured individual covariates and the distribution of the weights were highly similar between complete and incomplete cases. State-level weights were set to 1; patient weights were rescaled around the mean weight at each facility.[26]

See online supplementary figure 2 for all formulas. Stata V.14 was used for all analyses.

## Patient and public involvement

Patients nor the public were involved in the design, conduct or analysis of this study.

## RESULTS

A total of 27 338 patients were surveyed from 319 primary care facilities in 32 states; 25 745 of these provided complete data (94.2%, figure 2). The median age of respondents was 49 years, 71% were female and 70% had less than a high school education (table 1). A small minority of the visits represented preventive (8%) or dental (2%) care, versus curative (90%). The most common reasons for the visit were diseases of the circulatory system (20%) or endocrine, nutritional and metabolic diseases (19%). At the facility level, the average number of rooms in clinics was 20 (range=1–57), and the median number of enrolled patients was 83 691 (range=2356–468 618). A minority had nurses rooms attached (6%), the DIABETIMSS programme (19%) or nutritionists (39%). Participants lived in states where 41% of the population, on average, lived in poverty.

The majority of patients reported the staff greeted them by looking them in their eyes (82%), responded clearly to their questions (92%) and listened to them with attention and without interruptions (91%) (figure 3). Seventy-eight per cent reported experiencing all three of these

**Table 1** Patient, facility and state characteristics

| | n (%) |
|---|---|
| **Patient characteristics (N=25 745)** | |
| Sex | |
| Male | 7533 (29.3) |
| Female | 18 212 (70.7) |
| Age | |
| Median, range (IQR) | 49, 18–99 (34–63) |
| Level of education | |
| Less than primary school | 4006 (15.6) |
| Less than high school | 13 914 (54.0) |
| High school or more | 7825 (30.4) |
| Visit type | |
| Curative | 23 144 (89.9) |
| Preventive | 2024 (7.9) |
| Dental | 577 (2.2) |
| Reason for visit (ICD-10 code) | |
| Diseases of the circulatory system | 5132 (19.9) |
| Endocrine, nutritional and metabolic diseases | 4898 (19.0) |
| Other | 3085 (12.0) |
| Diseases of the respiratory system | 2386 (9.3) |
| Factors influencing health status and contact with health services | 1805 (7.0) |
| Pregnancy, childbirth and the puerperium | 1781 (6.9) |
| Diseases of the musculoskeletal system and connective tissue | 1710 (6.6) |
| Diseases of the digestive system | 1691 (6.6) |
| Symptoms, signs and abnormal clinical and laboratory findings, not elsewhere classified | 1664 (6.5) |
| Injury, poisoning and certain other consequences of external causes | 1593 (6.2) |
| **Facility characteristics (N=319)** | |
| No. of consultation rooms | |
| Mean (range), SD | 20.4 (1–57), 13.0 |
| Diabetes programme | |
| No diabetes programme | 258 (80.9) |
| Has diabetes programme | 61 (19.1) |

Continued

**Table 1** Continued

| | n (%) |
|---|---|
| Nutritionist | |
| No nutritionist | 194 (60.8) |
| At least one nutritionist | 125 (39.2) |
| Nurses room | |
| No nurse room | 300 (94.0) |
| At least one nurse room | 19 (6.0) |
| Patient population size | |
| Median (range), SD | 83 691 (2356–468 618), 86 231.0 |
| **State characteristics (N=32)** | |
| Percent poverty in state | |
| Mean (range), SD | 41.1 (20.4–76.2), 12.9 |

indicators of good verbal or non-verbal communication. Ninety-one per cent reported having the opportunity to share health concerns; 85% reported ending up with no doubts about their treatment. In 29% of visits, respondents reported excellent care overall (figure 4).

Examination of ICC and AUC values allowed us to consider general contextual effects of clustering at the facility and state levels. In terms of the ICC, the amount of individual variation in propensity to report positive experiences that was due to systematic differences between facilities rather than systematic differences between patients ranged from 6% for both verbal/non-verbal communication and opportunity to share health concerns to 12% for the global measure. Variation due to differences between states was 6% for the global measure and 3% for the other measures (table 2). The increase in AUC values between the single-level model with covariates and the model adding the facility random effect demonstrates that accounting for systematic differences between facilities increased predictive validity of the model by 8% to 12% (table 2); conversely, the addition of the random effect for state did not result in an increase in AUC values, suggesting that accounting for systematic differences between states did not improve the models' ability to correctly predict individual outcomes. Roughly two-thirds of individual outcomes were correctly predicted by the final models across the four outcomes.

Based on the PCV values (table 3), measured covariates at the facility level explained between 2% (global rating of visit) and 18% (opportunity to share health concerns) of the change in contextual variance. State poverty level explained a maximum of 3% of the change in variance. In the final multivariable models (Model 5 with state and facility covariates and random effects included), individuals within the same facility had higher odds of reporting positive communication if they were male (AOR 1.18, 95% CI 1.11 to 1.26), 49–62 or 63–99 years compared with 18–33 years of age (AOR 1.16 and 1.30, 95% CI 1.01 to 1.32 and 1.12 to 1.50, respectively), attended a visit related to pregnancy, childbirth or the postpartum period (AOR

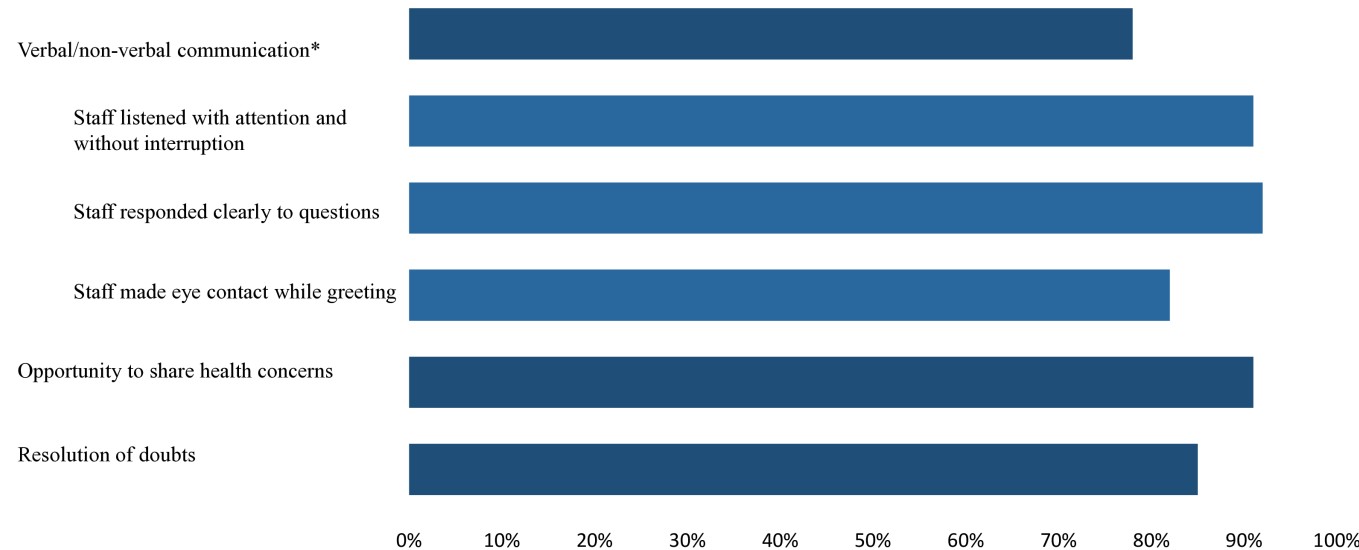

*Verbal/non-verbal communication is a composite variable comprised of those who reported positively to all three of the listed variables.

**Figure 3** Patient report of experience with communicating with healthcare providers (N=25 745).

1.30, 95% CI 1.02 to 1.67), or were seeking dental services (AOR 1.37, 95% CI 1.05 to 1.79) after adjusting for other individual, facility and state characteristics (table 3; see output from Models 1–4 in online supplementary table 2). More educated patients (with high school education or higher compared with less than primary school) within the same facility had lower odds of reporting high-quality communication, controlling for other covariates (AOR 0.84, 95% CI 0.76 to 0.94).

The pattern for other outcomes was slightly different. Individuals within the same facility had lower odds of reporting they were given the opportunity to share health concerns if they were 34–48, 49–62 and 63–99

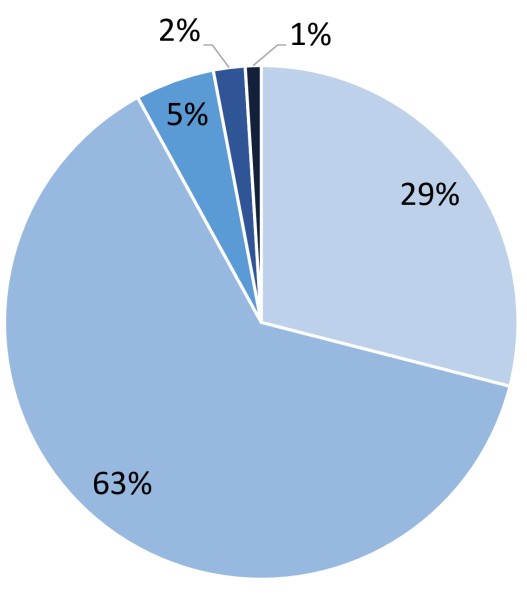

**Figure 4** Patient global rating of treatment received (N=25 745).

years compared with 18–33 years (AOR 0.83 and 0.83 and 0.82, 95% CI 0.72 to 0.95 and 0.72 to 0.97 and 0.70 to 0.96, respectively); being seen for visits categorised in ICD-10 codes as injury, poisoning and other consequences of external causes (AOR 0.62, 95% CI 0.50 to 0.77) or 'factors influencing health status or contact with health services' (0.38, 95% CI 0.30 to 0.49); or if they were being seen in a preventive care versus curative visit (AOR 0.27, 95% CI 0.24 to 0.30). Individuals within the same facility had lower odds of reporting resolution of doubts if they were being seen for symptoms, signs and abnormal clinical and laboratory findings, not elsewhere classified (AOR 0.58, 95% CI 0.48 to 0.72), being seen for injury, poisoning and other consequences of external causes (AOR 0.70, 95% CI 0.52 to 0.95), being seen for diseases of the musculoskeletal system and connective tissue (AOR 0.74, 95% CI 0.57 to 0.97), or being seen for other reason (AOR 0.69, 95% CI 0.57 to 0.83). Individuals within the same facility had higher odds of resolution of doubts if they had high school education or higher compared with less than primary school (AOR 1.18, 95% CI 1.05, 1.34) or were being seen for a preventive versus curative visit (AOR 1.26, 95% CI 1.05 to 1.50), controlling for other covariates. Finally, individuals within the same facility had higher odds of reporting overall excellent care if they were 49–62 and 63–99 years compared with 18–33 years (AOR 1.23 and 1.22, 95% CI 1.10 to 1.38 and 1.07 to 1.39, respectively) or if they had a primary school or high school education compared with less than primary school (AOR 1.13 and 1.35, 95% CI 1.01 to 1.26 and 1.16 to 1.57, respectively).

Across all facilities, larger patient population size affiliated with clinic was associated with lower odds of positive communication, opportunity to share concerns and resolution of doubts in the adjusted models (AORs comparing the smallest population size to the largest 0.32 and 0.43

**Table 2** General contextual effects

| | Positive verbal/non-verbal communication | Opportunity to share health concerns | Resolution of doubts | Excellent global experience rating |
|---|---|---|---|---|
| **Intraclass correlation (ICC)** | | | | |
| Addition of facility RE | | | | |
| Facility level coefficient | 0.06 | 0.05 | 0.08 | 0.10 |
| Addition of facility and state RE | | | | |
| Facility level coefficient | 0.06 | 0.06 | 0.08 | 0.12 |
| State level coefficient | 0.03 | 0.03 | 0.03 | 0.06 |
| **Area under curve (AUC)** | | | | |
| Single level with covariates | 0.56 (0.55–0.56) | 0.63 (0.61–0.64) | 0.56 (0.55–0.57) | 0.54 (0.53–0.55) |
| Addition of facility RE | 0.64 (0.63–0.65) | 0.71 (0.70–0.72) | 0.67 (0.66–0.68) | 0.66 (0.65–0.66) |
| Addition of facility and state RE | 0.64 (0.63–0.65) | 0.70 (0.69–0.71) | 0.67 (0.66–0.67) | 0.65 (0.65–0.66) |

RE, random effect.

and 0.48; 95% CI 0.13 to 0.74 and 0.21 to 0.90 and 0.25 to 0.93, respectively). The only facility-level IOR that did not include the null was patient population size (for all individual experience measures, but not the global measure), suggesting that this was the only measured facility-level characteristic whose association with patient experience was strong compared with remaining facility-level heterogeneity, conditional on state. For all other facility-level variables and for state poverty, IORs included the null.

## DISCUSSION

This novel analysis of patient experiences with primary care in Mexico identified generally positive perspectives on healthcare and found that up to 12% of variation in experience occurs at the facility level and up to 6% occurs at the state level. Models accounting for individual, facility and state characteristics correctly classified outcomes for two in three individuals, suggesting that these methods improve our insight on patient experience while pointing to the remaining unexplained variation. This suggests that quality improvement at the facility or regional level has the potential to improve individual experiences with care, although the degree of variation we uncovered at these higher levels is lower than that identified in other studies. For example, a study in general practice in Canada found that 20% of variance in perception of accessibility was attributable to practice-level characteristics,[15] and a patient survey among primary care patients in California, USA, found that 28%–48% was due to system-related factors.[27]

While smaller patient population size was clearly associated with better patient experience, only 2% to 18% of facility-level variation could be explained by measured covariates, suggesting further research is necessary in Mexico to identify levers for improvement. In the QUALICOPC study, conducted in 34—mainly European—countries, physicians providing a broader range of services and being paid through capitation had patients that reported more positive experiences, and more robust national primary care structure and higher health spending were also factors associated with better experience.[16] In-depth exploration of high-performing facilities in Mexico would help provide insights into these and other characteristics of facilities and the broader health system that may influence patient experience. Individual characteristics shaped experience ratings, reinforcing the importance of patients' expectations of care and the potential for differential treatment by providers to impact experience. Systematic assessment of patient expectations and provider bias may help to inform health system improvements.

The finding that larger patient population affiliated with clinic was correlated with worse reported patient experience on a number of dimensions is consistent with findings from other research[14] and suggests that particular attention may need to be paid to improving patient experience in settings with high patient volume.

The pattern of relationships between individual-level characteristics and patient experience was variable across outcomes. Men reported more positive communication than women; this may reflect a difference in how providers communicate with men or different expectations for care.[28] Older individuals tended to report more positive experiences overall and specifically with communication, consistent with a review of the impact of individual characteristics on experiences.[11] Older individuals were less likely than those between 18 and 33 years to report opportunity to share their health concerns, again either representing a difference in communication style on the part of providers or different expectations for care by patients by age. In terms of education, more educated individuals were less likely to report positive communication,

**Table 3** Multilevel logistic regression results: adjusted ORs (AORs) with 95% CI (N=25 745 patients, N=319 facilities, N=32 states)

| Individual level variables | Positive verbal/non-verbal communication AOR (95% CI) | Opportunity to share health concerns AOR (95% CI) | Resolution of doubts AOR (95% CI) | Excellent global experience rating AOR (95% CI) |
|---|---|---|---|---|
| Male sex | **1.18 (1.11 to 1.26)** | 1.06 (0.94 to 1.21) | 0.98 (0.87 to 1.12) | 1.07 (0.98 to 1.17) |
| Age | | | | |
| 18–33 years | Ref | Ref | Ref | Ref |
| 34–48 years | 1.08 (0.94 to 1.23) | **0.83 (0.72 to 0.95)** | 0.90 (0.79 to 1.02) | 1.04 (0.93 to 1.16) |
| 49–62 years | **1.16 (1.01 to 1.32)** | **0.83 (0.72 to 0.97)** | 0.95 (0.81 to 1.11) | **1.23 (1.10 to 1.38)** |
| 63–99 years | **1.30 (1.12 to 1.50)** | **0.82 (0.70 to 0.96)** | 0.98 (0.85 to 1.14) | **1.22 (1.07 to 1.39)** |
| Level of education | | | | |
| Less than primary school | Ref | Ref | Ref | Ref |
| Less than high school | 0.90 (0.80 to 1.02) | 1.01 (0.81 to 1.25) | 1.07 (0.96 to 1.19) | **1.13 (1.01 to 1.26)** |
| High school or more | **0.84 (0.76 to 0.94)** | 1.14 (0.92 to 1.41) | **1.18 (1.05 to 1.34)** | **1.35 (1.16 to 1.57)** |
| Visit type | | | | |
| Curative | Ref | Ref | Ref | Ref |
| Preventive | 1.02 (0.85 to 1.22) | **0.27 (0.24 to 0.30)** | **1.26 (1.05 to 1.50)** | 1.07 (0.88 to 1.30) |
| Dental | **1.37 (1.05 to 1.79)** | 1.09 (0.73 to 1.64) | 1.22 (0.89 to 1.68) | 1.21 (0.91 to 1.62) |
| Reason for visit | | | | |
| Diseases of the circulatory system | Ref | Ref | Ref | Ref |
| Endocrine, nutritional and metabolic diseases | 1.02 (0.88 to 1.18) | 1.01 (0.82 to 1.24) | 0.88 (0.73 to 1.06) | 0.95 (0.83 to 1.09) |
| Other | 0.89 (0.75 to 1.04) | .87 (0.72 to 1.05) | **0.69 (0.57 to 0.83)** | 0.95 (0.81 to 1.12) |
| Diseases of the respiratory system | 0.94 (0.82 to 1.08) | 1.16 (0.84 to 1.61) | .96 (0.76 to 1.23) | 0.95 (0.82 to 1.09) |
| Factors influencing health status and contact with health services | 1.12 (0.90 to 1.38) | **0.38 (0.30 to 0.49)** | 1.03 (0.82 to 1.30) | 1.14 (0.93 to 1.39) |
| Pregnancy, childbirth and the puerperium | **1.30 (1.02 to 1.67)** | 0.93 (0.73 to 1.20) | 0.89 (0.71 to 1.10) | 1.08 (0.88 to 1.34) |
| Diseases of the musculoskeletal system and connective tissue | 0.82 (0.66 to 1.02) | 0.82 (0.65 to 1.03) | **0.74 (0.57 to 0.97)** | 1.01 (0.89 to 1.20) |
| Diseases of the digestive system | 0.88 (0.73 to 1.06) | 0.86 (0.64 to 1.14) | 0.75 (0.55 to 1.03) | 0.86 (0.72 to 1.03) |
| Symptoms, signs and abnormal clinical and laboratory findings, not elsewhere classified | 0.89 (0.71 to 1.11) | 0.88 (0.67 to 1.17) | **0.58 (0.48 to 0.72)** | 1.04 (0.87 to 1.25) |
| Injury, poisoning and certain other consequences of external causes | 0.88 (0.73 to 1.06) | **0.62 (0.50 to 0.77)** | **0.70 (0.52 to 0.95)** | 0.90 (0.74 to 1.09) |
| **Facility level variables** | **AOR (95% CI) IOR** | **AOR (95% CI) IOR** | **AOR (95% CI) IOR** | **AOR (95% CI) IOR** |
| Number of consultation rooms | | | | |
| AOR | 1.65 (0.88 to 3.36) | 0.93 (0.37 to 2.34) | 0.85 (0.41 to 1.74) | 1.04 (0.51 to 2.10) |
| IOR | 0.92 to 2.95 | 0.50 to 1.72 | 0.45 to 1.60 | 0.43 to 2.49 |
| Presence of: | | | | |
| Diabetes programme | | | | |
| AOR | 1.02 (0.88 to 1.19) | 1.08 (0.81 to 1.45) | 1.11 (0.92 to 1.36) | 0.99 (0.82 to 1.19) |
| IOR | 0.57 to 1.83 | 0.59 to 2.00 | 0.59 to 2.10 | 0.41 to 2.38 |

**Table 3** Continued

| Facility level variables | AOR (95% CI) IOR | AOR (95% CI) IOR | AOR (95% CI) IOR | AOR (95% CI) IOR |
|---|---|---|---|---|
| Nutritionist | | | | |
| AOR | 1.00 (0.84 to 1.19) | 1.07 (0.83 to 1.40) | 0.95 (0.73 to 1.25) | 1.10 (0.89 to 1.35) |
| IOR | 0.56 to 1.56 | 0.58 to 1.99 | 0.51 to 1.79 | 0.46 to 2.63 |
| Nurses room | | | | |
| AOR | 0.87 (0.69 to 1.10) | 0.97 (0.67 to 1.40) | 0.98 (0.72 to 1.33) | 0.92 (0.79 to 1.08) |
| IOR | 0.49 to 1.56 | 0.52 to 1.79 | 0.52 to 1.84 | 0.38 to 2.21 |
| Patient population size | | | | |
| AOR | **0.32 (0.13 to 0.74)** | **0.43 (0.21 to 0.90)** | **0.48 (0.25 to 0.93)** | 1.19 (0.48 to 2.91) |
| IOR | 0.18 to 0.56 | 0.23 to 0.80 | 0.26 to 0.90 | 0.49 to 2.85 |
| **State level variable** | | | | |
| Percent poverty in state (minimum=20%; vs maximum=76%) | | | | |
| AOR | 1.12 (0.69 to 1.82) | 0.82 (0.50 to 1.35) | 0.48 (0.50 to 1.67) | 0.74 (0.42 to 1.30) |
| IOR | 0.62 to 2.02 | 0.46 to 1.47 | 0.42 to 1.97 | 0.33 to 1.67 |

AOR, adjusted OR; IOR, interval OR.

consistent with other research and likely due to higher expectations for care.[11] However, they were more likely to report resolution of doubts and positive overall experiences. This could represent the fact that those with the most education are empowered to be more 'active' participants in their care,[29] or possible provider discrimination against patients with less than primary school.

The reason for a patient's visit was consistently related to their experience, with preventive care visits eliciting higher ratings for resolution of doubts (although lower ratings for opportunity to share concerns), and dental visits being associated with better communication. This may reflect the different nature of preventive care, wherein the ability to resolve patients' doubts is more achievable than when patients are being seen for a particular medical concern. It could, however, reflect better quality care for patients in the preventive care or dental modules. Specific conditions were also associated with report of better or worse experiences in some cases, also likely reflecting either the nature of the visit or systematic differences in the ways providers treat patients who are being seen for different conditions.

These findings should be considered in light of several limitations. First, the data are cross-sectional and we are unable to draw causal inferences from the observed associations between independent and dependent variables. Further, although we were able to identify the overall amount of variance attributable to the facility and state levels, we had few potential explanatory factors available at those levels and could not explain the majority of the contextual variation. Second, available datasets did not indicate which providers patients had seen, keeping us from being able to parse the variance in patient experience attributable to differences between providers. Future efforts to collect national experience data in the IMSS system could benefit from collecting these data. Third, we did not have access to information on all IMSS primary care visits to be able to compare the demographics of our sample with the overall population of users. However, the response rate was high (87%), particularly compared with other patient experience surveys such as the Consumer Assessment of Healthcare Providers and Systems in the USA,[30] and the rate of missing data was fairly low (6%), suggesting reasonable representativeness to the population of IMSS users. Finally, the results are generalisable only to users of the IMSS system, as the survey did not include patients from other health sectors in Mexico.

Our findings shed light on the influences at individual, facility and state levels on patients' experiences with a large public health system in Mexico. These findings can inform efforts to improve patient experiences, thereby promoting confidence in the health system and better health outcomes. Key priorities for future work include further investigation of the facility-level factors accounting for differences in patient experience, as well as more robust monitoring of patients' expectations for care and their potential experiences with bias to better understand what accounts for individual level differences in experience.

**Author affiliations**
[1]Department of Family and Community Medicine, University of California, San Francisco, California, USA
[2]Epidemiology and Health Services Research Unit, CMN Siglo XXI, Mexican Institute of Social Security, Mexico City, Mexico
[3]Health Policy and Management, Harvard T.H. Chan School of Public Health, Boston, Massachusetts, United States
[4]Division of Social Protection and Health, Interamerican Development Bank, Kingston, Jamaica
[5]Global Health and Population, Harvard T. H. Chan School of Public Health, Boston, Massachusetts, USA

**Acknowledgements** The authors would like to acknowledge the generous help of Sarah Wulf in preparing the tables and figures for the manuscript.

**Contributors** KH, SVD and HHL conceived of the study; KH, HHL and DL conducted the analysis; KH drafted the manuscript; SVD, HHL, DL and RPC provided substantive feedback on the manuscript.

**Funding** HHL and DL received support from the Bill and Melinda Gates Foundation (Award OPP1161450, Kruk).

**Competing interests** None declared.

**Patient consent for publication** Not required.

**Ethics approval** This secondary analysis of deidentified data was approved by IMSS Research and Ethics Committee (No. R 2018-785-037).

**Provenance and peer review** Not commissioned; externally peer reviewed.

**Data availability statement** Patient-level and state-level data are available in public, open access repositories. Patient-level outcomes: http://datos.imss.gob. mx/. State-level poverty data: https://www.coneval.org.mx/coordinacion/entidades/ Paginas/inicioent.aspx. Facility-level data may be obtained from a third party and are not publicly available. Data on Mexican Institute of Social Security (IMSS) facility characteristics are available from IMSS on request after approval for analyses from the IMSS Health Research Ethics Committee (contact: comiteeticainv.imss@gmail. com).

**ORCID iDs**
Kelsey Holt http://orcid.org/0000-0001-9093-4322
Svetlana V Doubova http://orcid.org/0000-0002-0521-7095
Hannah H Leslie https://orcid.org/0000-0002-7464-3645

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
