## [Reviewer comments · BMJ Open]

ARTICLE DETAILS

TITLE (PROVISIONAL)	Factors associated with positive user experience with primary healthcare providers in Mexico: A multi-level modeling approach using national cross-sectional data
AUTHORS	Holt, Kelsey; Doubova, Svetlana; Lee, Dennis; Perez-Cuevas, Ricardo; Leslie, Hannah

VERSION 1 – REVIEW

REVIEWER	ANNA MARIA MURANTE SANT'ANNA SCHOOL - INSTITUTE OF MANAGEMENT - MANAGEMENT AND HEALTH LABORATORY
REVIEW RETURNED	02-Apr-2019

GENERAL COMMENTS	This manuscript analyses the predictors of patient experience with primary care services in Mexico. The analysis is based on secondary data collected at national level in 2016. The authors well define their research questions and describe method and results. They adopted a hierarchical model to catch out the multilevel effects of individuals and facilities' characteristics on patient experience. It is not clear for me if the facilities are 1 per region or more than 1 per region (I think they are more than 1 per region otherwise it means that in Mexico there are 319 regions. Is it right?). Hence, from a methodological point of you there is an important matter to overcome. The authors had to design a three-level model in order to include the regional level. This step is necessary due to the introduction of the indicator on the population's poverty that is measured at the regional level. This (necessary) improvement ensure a well defined univocal nested relationship of patients within facilities within regions. It is a mistake to include a regional characteristic at the facility level. This mistake alters the results on the ICC and, in turn, does not allow to correctly measure the variability at the facility level. The value reported by the authors is very high in comparison with those reported by other studies that analyzed the effect at the service/unit/provider level (usually lower than 10%). I conclude that it is necessary to re-perform the model after introducing this change. A last suggestion: please, read some papers on the QUALICOPC study. It was inducted in more than 30 countries in Europe (also middle income countries like Bulgaria, Macedonia, Turkey) and included analyses on patient experience with primary care. Also multilevel analyses were conducted.
---

REVIEWER	Justine Strand de Oliveira
-----------------	----------------------------

	Duke University, Family Medicine and Community Health
REVIEW RETURNED	06-May-2019

GENERAL COMMENTS	Novel study which provides useful insight into the experience of primary health care in a middle income country. A worthwhile addition to the literature.
---

VERSION 1 – AUTHOR RESPONSE

Reviewer(s)' Comments to Author:

Reviewer: 1

Reviewer Name: ANNA MARIA MURANTE

Institution and Country: SANT'ANNA SCHOOL - INSTITUTE OF MANAGEMENT - MANAGEMENT AND HEALTH LABORATORY Please state any competing interests or state 'None declared': None declared

Please leave your comments for the authors below This manuscript analyses the predictors of patient experience with primary care services in Mexico. The analysis is based on secondary data collected at national level in 2016.

The authors well define their research questions and describe method and results. They adopted a hierarchical model to catch out the multilevel effects of individuals and facilities' characteristics on patient experience.

It is not clear for me if the facilities are 1 per region or more than 1 per region (I think they are more than 1 per region otherwise it means that in Mexico there are 319 regions. Is it right?). Hence, from a methodological point of you there is an important matter to overcome. The authors had to design a three-level model in order to include the regional level. This step is necessary due to the introduction of the indicator on the population's poverty that is measured at the regional level. This (necessary) improvement ensure a well defined univocal nested relationship of patients within facilities within regions. It is a mistake to include a regional characteristic at the facility level. This mistake alters the results on the ICC and, in turn, does not allow to correctly measure the variability at the facility level. The value reported by the authors is very high in comparison with those reported by other studies that analyzed the effect at the service/unit/provider level (usually lower than 10%).

I conclude that it is necessary to re-perform the model after introducing this change.

- Thank you very much for sharing your expert recommendation. You are correct that we have multiple facilities per state. We have followed your suggestion and revamped the model to include a third level for state. Our methods, results and discussion sections and table 2 have been adjusted accordingly.

A last suggestion: please, read some papers on the QUALICOPC study. It was inducted in more than 30 countries in Europe (also middle income countries like Bulgaria, Macedonia, Turkey) and included analyses on patient experience with primary care. Also multilevel analyses were conducted.

- Thank you for bringing this study to our attention. We located the recent article published in July 2019 synthesizing results of the study and have added mention of the findings related to facility and systems level factors associated with positive patient experience in the Introduction and Discussion section.

Reviewer: 2

Reviewer Name: Justine Strand de Oliveira Institution and Country: Department of Family Medicine and Community Health, Duke University, United States Please state any competing interests or state 'None declared': None declared

Please leave your comments for the authors below Novel study which provides useful insight into the experience of primary health care in a middle income country. A worthwhile addition to the literature.
 • Thank you for this review.

VERSION 2 – REVIEW

REVIEWER	Juan Merlo Lund University, Faculty of Medicine, Social Epidemiology
REVIEW RETURNED	18-Oct-2019

GENERAL COMMENTS	The authors state the objective of their paper is to investigate factors associated with patient experience with primary care in a large public health system in Mexico. They perform a multilevel logistic regression model to analyse a cross-sectional 2016 national satisfaction survey conducted by the Mexican Social Security Institute with beneficiaries over age 18. Patient data (level 1) were merged with facility data (level 2) and information on poverty by state (level 3). The multilevel analysis focuses on measures of association (odds ratios) and measures of variance like the intraclass correlation coefficient (ICC) and the median odds ratio (MOR). They also consider the cluster specific nature of the multilevel modelling and express higher level association as interval odds ratio (IOR). They conclude that primary care quality improvement might bolster individual experiences with care, given that up to 12% of the variation in experience was attributable to facility and state level differences. The relationship between individual characteristics and experience ratings reinforces the importance of patients' expectations of care and the potential for differential treatment by providers to impact experience. The study is worthy and relevant. However, I have several comments, some of them major, that should be considered to improve this work. As the authors state that the findings are not generalizable to other sectors. However, with some improvements this kind of analytical approach should be standard when it comes to the quantitative assessment of health care quality. This study has two components. One is the substantive question concerning the investigation of factors associated with patient experience with primary care in Mexico. The other is the applied methodology itself. From the methodological perspective I think the paper will benefit if the authors consider the work performed using a similar multilevel modelling approach for investigating health care quality and geographical differences. See for instance the following publications (1) (2) (3) (4) A relevant idea the authors should use is the distinction between measures of association expressed as Specific Contextual effects
---

(SCE) and measures of variance expressed as General Contextual Effects (GCE)
For instance, the objective of the study is restricted to measures of association (i.e., “to investigate factors associated with patient experience with primary care in a large public health system in Mexico”). However, the conclusions are based on measures of variance (i.e., “primary care quality improvement might bolster individual experiences with care, given that up to 12% of the variation in experience was attributable to facility and state level differences”).

A general comment

The authors evaluate the share of the individual variation that is at the facility and state levels. This share expresses the general (unspecified) contextual effect (GCE) of those levels. That is the degree of clustering of patient satisfaction without considering any contextual characteristic other than the name of the facility-state. In other words, the same patient would have a different satisfaction if he/she was treated in another facility-state even if we do not know the reasons for it.

In addition they present specific contextual variables at both levels to observe their association with patient satisfaction (SCE). This approach tries to understand the specific mechanism that explains the GCE.

However, as it now, the authors perform models that include all variables together and the implicit concepts are unclearly defined. Besides, the model discourse is clearly insufficient for the study. The authors must perform a series of analyses to clarify the research question. I suggest the following steps (see also (1))

- 1.- identify the individual variables that could confound the differences between facilities and states
- 2.- Perform a conventional logistic regression including the individual variables
 - Give the OR (95% CI)
- 3.- Add a random effect for the facility and for the state
 - Give the OR (95% CI) for the individual variables
 - Give the facility and state variances. Calculate the ICC for those levels (this is the “ceiling” GCE) and the MOR (if you like it)
- 4.- Add fixed effects for the specific characteristics of the levels (SCE) and provide the population average OR (95% CI) and the 80% IOR

Here you can also provide the proportional change of the variance (PCV) from step 3 to step 4. In this way you know the proportion of the contextual variance that is explained by the specific contextual characteristics.

You can also add steps including first a random effect for the facility and later a random effect for the state.

The advantage of this modelling structure is that you avoid the problem of rescaling in multilevel logistic regression when you add individual variables to a multilevel model.

You should also add information on the area under the ROC curve (AUC) for all the models (see also (1))

Following this model logic, we could understand the discriminatory accuracy of the individual characteristics in relation to patient satisfaction, the GCE of the context and the added discriminatory accuracy when adding the levels as well as possible SCE behind

the GCE. This procedure will better support your conclusions. In the present form the information is not properly used. As it is now in the tables ICC are adjusted for contextual characteristics which does not inform of the initial CGE (step 3 above). You inform in the text (page 8, lines 16-22) that the ICC does not change so much after adjustment for both individual and contextual characteristics, but the meaning of this finding will be better captured using the model procedure I propose. Also, in page 9, lines 35-38 you state "While smaller patient population size was clearly associated with better patient experience, much of the facility-level variation could not be explained by measured covariates, suggesting further research is necessary. Again, I cannot see this information in the tables and the meaning of this finding will be better captured using the model procedure I propose."

Besides you write about the component of the total variance shared at the facility and at state. Therefore, you could use the concept of variance partition coefficient (VPC) even if it is analogous to the ICC for perfect hierarchical structures

Other comments

Please provide a flow chart indicate the participation rate (87%) and initial number of patients, the reasons and the number of patients that were dropped from the study and those with missing values. Indicate very clearly the number of facilities and states as well the number (min - max) of patient per contextual unit.

The percentage of missing values is about 26% (1593/25745) which needs to be considered. However, rather than weighting I would prefer to read a description of the differences between people with and without missing and a discussion on how missing could affect the results. Inform, if possible, on the differences between respondents and non-respondents and a discussion on how the sample is selected by response.

In the Appendix provide the equations for the different models. It is unclear which models you have performed in addition to those included in the tables.

This is a nice study initiative and I hope my comments help you.

REFERENCES

1. Merlo J, Wagner P, Ghith N, Leckie G. An Original Stepwise Multilevel Logistic Regression Analysis of Discriminatory Accuracy: The Case of Neighbourhoods and Health. *PLoS One*. 2016;11(4):e0153778.
2. Merlo J, Viciano-Fernandez FJ, Ramiro-Farinas D, LDAP RG. Bringing the individual back to small-area variation studies: A multilevel analysis of all-cause mortality in Andalusia, Spain. *Social Science & Medicine*. 2012;75(8):1477-87.
3. Ohlsson H, Lindblad U, Lithman T, Ericsson B, Gerdtham UG, Melander A, et al. Understanding adherence to official guidelines on statin prescribing in primary health care--a multi-level methodological approach. *Eur J Clin Pharmacol*. 2005;61(9):657-65.

	4. Ghith N, Frolich A, Merlo J. The role of the clinical departments for understanding patient heterogeneity in one-year mortality after a diagnosis of heart failure: A multilevel analysis of individual heterogeneity for profiling provider outcomes. PLoS One. 2017;12(12):e0189050.
--	---

VERSION 2 – AUTHOR RESPONSE

The authors state the objective of their paper is to investigate factors associated with patient experience with primary care in a large public health system in Mexico. They perform multilevel logistic regression model to analyse a cross-sectional 2016 national satisfaction survey conducted by the Mexican Social Security Institute with beneficiaries over age 18. Patient data (level 1) were merged with facility data (level 2) and information on poverty by state (level 3).

The multilevel analysis focuses on measures of association (odds ratios) and measures of variance like the intraclass correlation coefficient (ICC) and the median odds ratio (MOR). They also consider the cluster specific nature of the multilevel modelling and express higher level association as interval odds ratio (IOR).

They conclude that primary care quality improvement might bolster individual experiences with care, given that up to 12% of the variation in experience was attributable to facility and state level differences. The relationship between individual characteristics and experience ratings reinforces the importance of patients' expectations of care and the potential for differential treatment by providers to impact experience.

The study is worthy and relevant. However, I have several comments, some of them major, that should be considered to improve this work.

As the authors state that the findings are not generalizable to other sectors. However, with some improvements this kind of analytical approach should be standard when it comes to the quantitative assessment of health care quality.

This study has two components. One is the substantive question concerning the investigation of factors associated with patient experience with primary care in Mexico. The other is the applied methodology itself.

From the methodological perspective I think the paper will benefit if the authors consider the work performed using a similar multilevel modelling approach for investigating health care quality and geographical differences. See for instance the following publications (1) (2) (3) (4)

Thank you for providing these references, which we have reviewed closely. As we describe in more detail below, we have adjusted our modeling approach to fit more closely with your recommendation.

A relevant idea the authors should use is the distinction between measures of association expressed as Specific Contextual effects (SCE) and measures of variance expressed as General Contextual Effects (GCE) For instance, the objective of the study is restricted to measures of association (i.e., "to investigate factors associated with patient experience with primary care in a large public health system in Mexico"). However, the conclusions are based on measures of variance (i.e., "primary care

quality improvement might bolster individual experiences with care, given that up to 12% of the variation in experience was attributable to facility and state level differences”).

A general comment

The authors evaluate the share of the individual variation that is at the facility and state levels. This share expresses the general (unspecified) contextual effect (GCE) of those levels. That is the degree of clustering of patient satisfaction without considering any contextual characteristic other than the name of the facility-state. In other words, the same patient would have a different satisfaction if he/she was treated in another facility-state even if we do not know the reasons for it.

In addition, they present specific contextual variables at both levels to observe their association with patient satisfaction (SCE). This approach tries to understand the specific mechanism that explains the GCE.

However, as it now stands, the authors perform models that include all variables together and the implicit concepts are unclearly defined.

Thank you for pointing out that we had not previously made clear our intention to explore both general and specific contextual effects. We have revised our stated objectives in the Introduction section, clarified in Methods how analyses contribute to examining general and specific contextual effects, and modified the discussion of our findings, to reflect this dual objective. In addition, we have revised our modeling strategy, as described in detail below.

Besides, the model discourse is clearly insufficient for the study.

The authors must perform a series of analyses to clarify the research question. I suggest the following steps (see also (1))

- 1.- identify the individual variables that could confound the differences between facilities and states
- 2.- Perform a conventional logistic regression including the individual variables

Give the OR (95% CI)

- 3.- Add random effects for the facility and for the state

Give the OR (95% CI) for the individual variables

Give the facility and state variances. Calculate the ICC for those levels (this is the “ceiling” GCE) and the MOR (if you like it)

- 4.- Add fixed effects for the specific characteristics of the levels (SCE) and provide the population average OR (95% CI) and the 80% IOR. Here you can also provide the proportional change of the variance (PCV) from step 3 to step 4. In this way you know the proportion of the contextual variance that is explained by the specific contextual characteristics.

You can also add steps including first a random effect for the facility and later a random effect for the state.

The advantage of this modelling structure is that you avoid the problem of rescaling in multilevel logistic regression when you add individual variables to a multilevel model.

You should also add information on the area under the ROC curve (**AUC**) for all the models (see also (1))

Following this model logic, we could understand the discriminatory accuracy of the individual characteristics in relation to patient satisfaction, the GCE of the context and the added discriminatory accuracy when adding the levels as well as possible SCE behind the GCE. This procedure will better support your conclusions. In the present form the information is not properly used. As it is now in the tables ICC are adjusted for contextual characteristics which does not inform of the initial CGE (step 3 above). You inform in the text (page 8, lines 16-22) that the ICC does not change so much after adjustment for both individual and contextual characteristics, but the meaning of this finding will be better captured using the model procedure I propose.

Also, in page 9, lines 35-38 you state "While smaller patient population size was clearly associated with better patient experience, much of the facility-level variation could not be explained by measured covariates, suggesting further research is necessary." Again, I cannot see this information in the tables and the meaning of this finding will be better captured using the model procedure I propose.

Besides you write about the component of the total variance shared at the facility and at state. Therefore, you could use the concept of variance partition coefficient (VPC) even if it is analogous to the ICC for perfect hierarchical structures

We have followed the suggestion to re-do the modeling. Our steps are summarized below, and the Methods, Results, and Discussion have been updated accordingly.

- 1) Flat model with individual covariates (shown in Supplement), calculate AUC
- 2) Add RE for facility (shown in Supplement), calculate ICC, AUC
- 3) Add facility-level effects, calculate population average ORs (shown in Supplement), PCV
- 4) Add RE for state (shown in Supplement), calculate ICC, AUC
- 5) Add state poverty level, calculate population average ORs (Final model displayed in Table 3 of paper), PCV and IORs

Given our objective to assess multiple outcomes on patient perspectives, we reviewed the potential measures to include and their prior reporting in existing literature. We selected those we determined provided the most insight describing the specific and general contextual effects; we elected not to further complexify the results with the MOR and VPC.

Other comments

Please provide a flow chart indicate the participation rate (87%) and initial number of patients, the reasons and the number of patients that were dropped from the study and those with missing values. Indicate very clearly the number of facilities and states as well the number (min - max) of patient per contextual unit.

Thank you for this suggestion; we have added the requested Figure

The percentage of missing values is about 26% (1593/25745) which needs to be considered. However, rather than weighting I would prefer to read a description of the differences between people with and without missing and a discussion on how missing could affect the results. Inform, if possible, on the differences between respondents and non-respondents and a discussion on how the sample is selected by response.

The amount of missing data we had was 6%, not 26%; we employed inverse probability of censoring weighting as a more robust approach than complete case analysis, which requires the implausible assumption of missingness completely at random. We have added details on covariate balance to the supplement and mentioned the response rate and missingness rate more explicitly in the Discussion.

In the Appendix provide the equations for the different models. It is unclear which models you have performed in addition those included in the tables.

Thanks, we have updated the formulas in the Supplement accordingly

This is a nice study initiative and I hope my comments help you.

REFERENCES

1. Merlo J, Wagner P, Ghith N, Leckie G. An Original Stepwise Multilevel Logistic Regression Analysis of Discriminatory Accuracy: The Case of Neighbourhoods and Health. PLoS One. 2016;11(4):e0153778.
 2. Merlo J, Viciano-Fernandez FJ, Ramiro-Farinas D, LDAP RG. Bringing the individual back to small-area variation studies: A multilevel analysis of all-cause mortality in Andalusia, Spain. Social Science & Medicine. 2012;75(8):1477-87.
 3. Ohlsson H, Lindblad U, Lithman T, Ericsson B, Gerdtham UG, Melander A, et al. Understanding adherence to official guidelines on statin prescribing in primary health care--a multi-level methodological approach. Eur J Clin Pharmacol. 2005;61(9):657-65.
 4. Ghith N, Frolich A, Merlo J. The role of the clinical departments for understanding patient heterogeneity in one-year mortality after a diagnosis of heart failure: A multilevel analysis of individual heterogeneity for profiling provider outcomes. PLoS One. 2017;12(12):e0189050.
- <Comments Merlo.pdf><IMSS patient satisfaction 1aug19_clean.docx>

VERSION 3 – REVIEW

REVIEWER	Juan Merlo Faculty of Medicine, Lund University, Sweden
REVIEW RETURNED	18-Dec-2019
GENERAL COMMENTS	The authors have done a suitable work. (Please, spell out the acronym AOR the first time)